**Data Availability Statement:** All relevant data are within the manuscript and its Supporting Information files.

# Full-field optical coherence tomography for the diagnosis of giant cell arteritis

**Thomas Maldiney**[1☯], **Hélène Greigert**[1,2☯], **Laurent Martin**[2,3], **Emilie Benoit**[4], **Catherine Creuzot-Garcher**[5], **Pierre-Henry Gabrielle**[5], **Jean-Marie Chassot**[6], **Claude Boccara**[4,6], **Daniel Balvay**[7], **Bertrand Tavitian**[7,8], **Olivier Clément**[7,8], **Sylvain Audia**[1,2], **Bernard Bonnotte**[1,2‡], **Maxime Samson**[1,2‡]*

1 Department of Internal Medicine and Clinical Immunology, François Mitterrand Hospital, Dijon University Hospital, Dijon, Bourgogne-Franche-Comté, France, 2 University of Bourgogne-Franche-Comté, INSERM, EFS BFC, UMR1098, RIGHT Interactions Greffon-Hôte-Tumeur/Ingénierie Cellulaire et Génique, Dijon, Bourgogne-Franche-Comté, France, 3 Department of Pathology, François Mitterrand Hospital, Dijon University Hospital, Dijon, Bourgogne-Franche-Comté, France, 4 LLTech SAS, Paris, France, 5 Department of Ophthalmology, François Mitterrand Hospital, Dijon University Hospital, Dijon, Bourgogne-Franche-Comté, France, 6 Institut Langevin, ESPCI Paris, CNRS, PSL University, Paris, France, 7 Université de Paris, PARCC, INSERM, Paris, France, 8 Radiology Department, Hôpital Européen Georges Pompidou, Université Paris Descartes Sorbonne Paris Cité, Assistance Publique-Hôpitaux de Paris, Paris, France

☯ These authors contributed equally to this work.
‡ These authors also contributed equally to this work.
* maxime.samson@u-bourgogne.fr

## Abstract

Histopathological examination of temporal artery biopsy (TAB) remains the gold standard for the diagnosis of giant cell arteritis (GCA) but is associated with essential limitations that emphasize the need for an upgraded pathological process. This study pioneered the use of full-field optical coherence tomography (FF-OCT) for rapid and automated on-site pathological diagnosis of GCA. Sixteen TABs (12 negative and 4 positive for GCA) were selected according to major histopathological criteria of GCA following hematoxylin-eosin-saffron-staining for subsequent acquisition with FF-OCT to compare structural modifications of the artery cell wall and thickness of each tunica. Gabor filtering of FF-OCT images was then used to compute TAB orientation maps and validate a potential automated analysis of TAB sections. FF-OCT allowed both qualitative and quantitative visualization of the main structures of the temporal artery wall, from the internal elastic lamina to the *vasa vasorum* and red blood cells, unveiling a significant correlation with conventional histology. FF-OCT imaging of GCA TABs revealed destruction of the media with distinct remodeling of the whole arterial wall into a denser reticular fibrous neo-intima, which is distinctive of GCA pathogenesis and accessible through automated Gabor filtering. Rapid on-site FF-OCT TAB acquisition makes it possible to identify some characteristic pathological lesions of GCA within a few minutes, paving the way for potential machine intelligence-based or even non-invasive diagnosis of GCA.

 

**Funding:** This work was supported by grants from the 'Fondation ARTHRITIS (2017-2018)' and the 'Groupement Interrégional de Recherche Clinique et d'Innovation Est (GIRCI) - Appel à Projet Jeunes Chercheurs' 2013 and 2014 (Clinicaltrials.gov NCT02158208 and NCT02857192). The funder did not provide any salary for authors, had no role in the study design, data collection and analysis, decision to publish, or preparation of the manuscript. The specific roles of these authors are articulated in the 'author contributions' section. LLTech SAS did not play any role as a funding organization for this study.

**Competing interests:** We declare the paid employment of LLTech SAS (E. B.) to provide additional FF-OCT images acquisition and analysis. Besides, the commercial affiliation with LLTech SAS does not alter our adherence to PLOS ONE policies on sharing data and materials.

## Introduction

Giant cell arteritis (GCA) is a large vessel vasculitis that mainly affects the aorta and the branches of the external carotid, with a predilection for the temporal arteries [1]. Even though we now have an accurate understanding of its complex pathogenesis, the causative agent of GCA is still unknown [2]. Mostly occurring in northern European females between 50 and 80 years old, the predominant cranial phenotype is usually revealed by new-onset headache, temporal artery tenderness, jaw claudication, and partial or complete visual loss associated with possible systemic symptoms, notably fever, weight loss and weakness [3]. The critical complications of GCA include anterior ischemic optic neuropathy, stroke, aortic aneurysm or dissection; these serious complications being responsible for the prognosis of the disease and the need for prolonged high-dose glucocorticoid treatment [4].

The diagnosis of GCA usually relies on the association of concurrent clinical, biological and pathological features of vasculitis that are revealed by temporal artery biopsy (TAB) [5]. Significant advances in the field of medical imaging have improved the assessment of the extent of vasculitis and refined non-invasive diagnosis and follow-up [6,7]. For instance, the validity of hypoechoic thickening surrounding the temporal artery wall with color duplex sonography (CDS), also referred to as the halo sign, was confirmed at least three times in a meta-analysis for the diagnosis and follow-up of GCA [8]. However, the combination of intense infiltration of mononuclear cells in the three layers of the artery, fragmentation of the internal elastic lamina (IEL), intimal hyperplasia and neoangiogenesis on TAB histological examination undoubtedly remains the gold standard for GCA diagnosis in all study group guidelines [9,10].

Apart from rare local complications [11], TAB is a safe procedure [12]. Nevertheless, the segmental and focal nature of transmural inflammation in GCA generates skip lesions [13] and is responsible for a significant false-negative rate of up to 30% [14] that makes it necessary to either increase biopsy length [15] or to perform a contralateral TAB [16]. These limitations emphasize the potential interest and need for an upgraded pathological procedure dedicated to the diagnosis of GCA.

Based upon an optimization of the technology described by Fujimoto and colleagues in the early 1990s [17,18], full-field optical coherence tomography (FF-OCT) exploits *en face* white-light interference microscopy to provide not only ultra-high resolution images of biological structures [19] but also subcellular metabolic contrast in the tissue depth [20]. When compared to other modalities such as conventional OCT or even confocal microscopy, FF-OCT was demonstrated to significantly improve spatial resolution by a factor varying from five to ten depending on the acquisition axis [21]. Until now, most groups have focused on the potential role of FF-OCT during oncologic interventions as new routine approach to surgical pathology [22], and, except for one preliminary study in which the superficial temporal arteries were imaged with dermal OCT [23], there has been no reported attempt to employ high definition interference microscopy for the pathological diagnosis of GCA. The present work pursues the hypothesis that FF-OCT could help both the clinician and pathologist to improve TAB performance, and compares, for the first time, FF-OCT and conventional histological examination for the pathological diagnosis of GCA.

## Materials and methods

### Ethics statement

Patients included in this study participated in two studies involving GCA patients (Clinicaltrials.gov NCT02158208 and NCT02857192) and gave both oral and written informed consent for the use of their temporal arteries for subsequent research in the field of GCA. This study

was approved by the Institutional Review Board and the Ethics Committee of the Dijon University Hospital.

## Preparation of TAB sections

All patients suspected of GCA and scheduled for TAB surgery at the Dijon University Hospital Ophthalmology department from January 2013 to December 2016 were included. TAB was performed according to standard procedure, and fresh biopsies were sent to the pathology department. A ten millimeter segment was used for conventional hematoxylin-eosin-saffron (HES) staining, and the other part of the artery segment was immediately frozen at -196˚C in fetal bovine serum and dimethyl sulfoxide (10%). The day of FF-OCT imaging, samples were slowly defrosted. The surrounding tissue was removed, and transversal 1 mm-thick sections were cut with a triangular-bladed scalped and placed in complete RPMI culture medium before placement on the sample holder for image acquisition.

## Histological TAB selection

A total of sixteen TABs were selected for subsequent analysis with optical coherence microscopy. Twelve negative TABs were identified according to the absence of mononuclear cell infiltrate, IEL fragmentation or neoangiogenesis and defined as the control TABs. In these control samples, the pathologist studied the qualitative aspect of the temporal artery wall to distinguish between negative TAB with normal intima (thinner than media, referred to as niTAB.1 to 9, n = 9) and negative TAB with intimal hyperplasia (thicker than media, referred to as ihTAB.1 to 3, n = 3). These control TABs were compared to four specimens that met the major histopathological criteria for the diagnosis of GCA (referred to as gcaTAB.1 to 4, n = 4).

## FF-OCT imaging

FF-OCT images were acquired with a commercially available FF-OCT apparatus (Light-CTScanner, LLTech SAS, Paris, France) [24]. Briefly, illumination was provided by a LED source with short coherence length ensuring a sectioning ability or axial resolution of 1 μm. In the FF-OCT set-up, 10x microscope objectives are placed in the interferometer arms in the Linnik configuration, bringing a transverse resolution of 1.5 μm. Following full-field illumination of the axial TAB section, FF-OCT images were captured with a complementary metal oxide semiconductor camera. The theoretical penetration depth for the TAB specimen was approximately 200 μm. The TAB section was placed in the dedicated sample holder with its revolution axis perpendicular to the imaging plane so that one FF-OCT slice showed the architecture of the TAB section from the lumen to the outer wall. A series of FF-OCT slices with 1.5 μm spacing were recorded in depth, and ImageJ 1.52o software was used for axial reconstruction of TAB FF-OCT imaging following z-stacking of a minimum of 20 images.

## Image and statistical analysis

Quantitative FF-OCT image analysis and tunica thickness were accessible with a contrast-based ImageJ 1.52o protocol (Plot Profile Function) and calculated as the mean of three representative measurements throughout each TAB section. NDP.view software version 2.6.17, provided by Hamamatsu, allowed similar measurements from scanned glass slides following HES staining. Statistics were calculated using GraphPad Prism version 5. For intima-to-media ratios (I/M), values reported as medians and interquartile ranges were discriminated by Mann-Whitney tests. The Pearson r coefficient was calculated to evaluate the strength of the linear correlation between histology and FF-OCT measurements of media or intima

thicknesses. Interval two-tailed P < 0.05 was considered statistically significant. Finally, orientation maps were calculated for a selection of both healthy and GCA-positive TABs following Gabor filtering of the axial reconstructed images with a custom-made software based on Matlab 2018b (Matworks, Natick, MA).

## Results

### Qualitative FF-OCT imaging

Similar to histological preparation, TAB sections acquired with FF-OCT allow the identification of several important structures within the artery wall (Fig 1). Notably, the tripartite architecture is perceived with a clear separation between intima, media and adventitia (Fig 1A). Interestingly, the physical junction between the intima and media appears as a thin hypo-reflective serpentine band that most obviously corresponds to the IEL (Fig 1A, black arrow). Moreover, in Fig 1B and 1C obtained with FF-OCT, the *vasa vasorum* can be seen distinctly within the arterial wall and the red blood cells can be identified precisely, returning a spherical contrast highly similar to the one obtained with conventional histology (Fig 1D to 1F). Indeed, the *vasa vasorum* display similar architecture with both techniques, revealing small (20 to 80 μm) blood vessels characterized by thin elastic walls and a round to oval shape directly inserted into the outlying thread of the temporal artery (*i.e.* mostly between the media and the adventitia layers) (Fig 1B and 1E, white arrow shows arterial thrombi). Magnification of the lumen of the *vasa vasorum* makes it possible to observe the red blood cells. These cells resemble partially transparent pink-colored ovoid structures following HES staining or a collection of iso-reflective dots with a surrounding hypo-reflective annulus on direct FF-OCT acquisition (Fig 1C and 1F, black asterisk).

FF-OCT acquisition and histological images from negative TAB samples are compared in Fig 2A to 2D. Fig 2A and 2B display a representative negative TAB specimen with a thin intimal layer (niTAB). Fig 2C and 2D show a negative TAB with intimal hyperplasia (hiTAB). No matter the group of negative TAB, the overall architecture of the vessels is preserved, and there is a clear distinction between intima, media and adventitia. Indeed, the tunica media displays a relative hyper-reflectivity on the image acquired with FF-OCT when compared with the tunica intima, whose thin muscle fibers mostly run parallel to the global circular orientation of the TAB section (see magnified region from Fig 2A and 2C). Similar conclusions regarding the differential contrast and circular symmetry within the two inner layers of the arterial wall

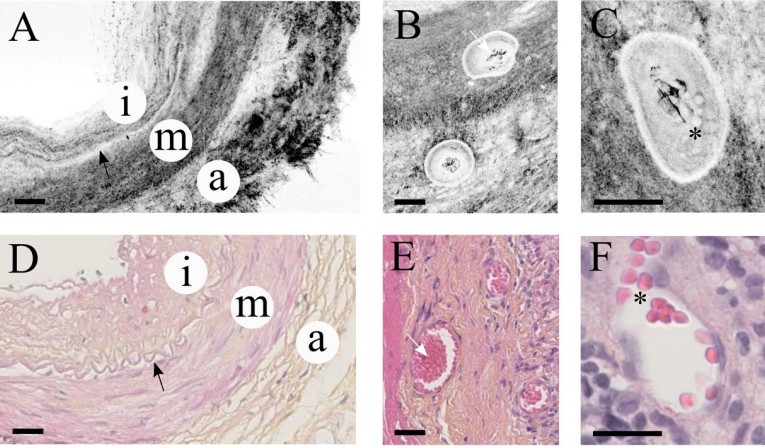

**Fig 1.**

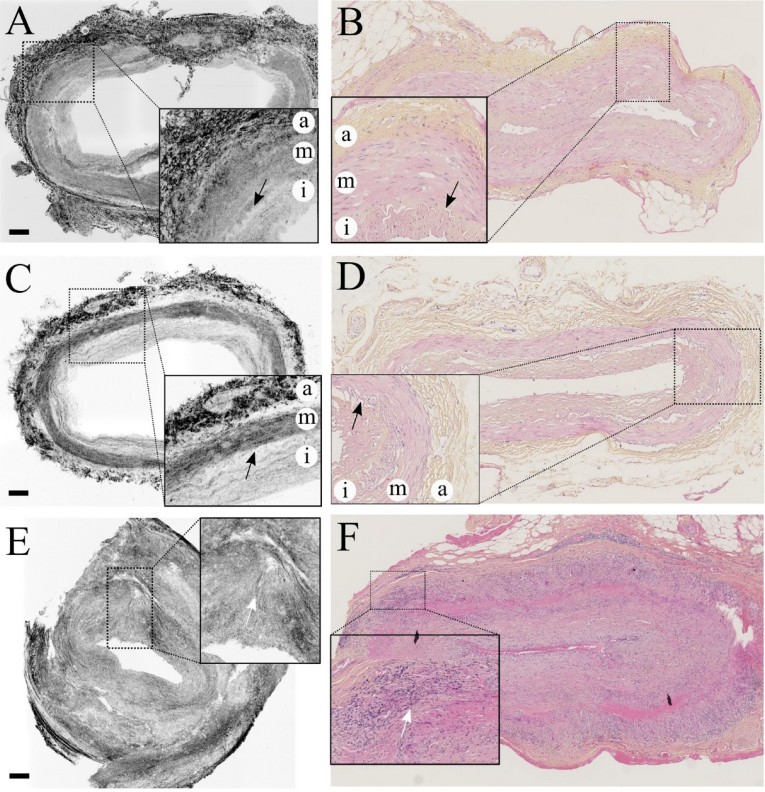

**Fig 2.**

separated by the IEL, which appears as a hypo-reflective strip in FF-OCT, can be drawn from the analysis of all negative TAB specimens (S1 and S2 Figs). In Fig 2A and 2C we can see that the tunica adventitia is constructed on a denser and more complex fibrous connective tissue that also seems to follow the overall circular symmetry of the system. When compared with Fig 2A, the FF-OCT-acquired TAB section from Fig 2C is characterized by an increased intima thickness (see magnified region). These observations can mostly be transposed for the comparison of all hiTAB specimens detailed in S1 and S2 Figs. The results obtained with FF-OCT analysis largely correlate with the data obtained after conventional histology (Fig 2B and 2D). Indeed, the intima in Fig 2B appears much thinner than in Fig 2D in which the intima is thicker than the media.

In Fig 2E and 2F and S3 Fig, the TABs are positive for GCA. The conventional histopathological images show a relatively preserved media that is strongly infiltrated by T-cells, macrophages and multinucleated cells (see the magnified region from Fig 2F). By contrast, FF-OCT acquisition demonstrates a complete disruption of both regular reflectivity and circularity of the media and intima-associated connective tissue fibers due to the infiltration of inflammatory cells. This process remodels the structure of the artery into a denser, reticular, fibrous, collagen-rich structure responsible for both the progressive destruction of the media and the formation of a neo-intima (see the magnified region from Fig 2E). Fig 2E, obtained with FF-OCT, confirms the fragmentation of the internal elastic lamina along with a rebalancing of the contrasts throughout the netlike fibrous structure connecting all three layers. Similar to the corresponding image obtained with conventional histology (Fig 2F), there is no clear distinction between the intima and the media, which is consistent with the stage of the disease. The

same conclusion can be drawn from S3 Fig, which shows the supporting material in which the reticular fibrous neo-intima almost completely obstructs the arterial lumen.

## Quantitative FF-OCT imaging

Given that FF-OCT images provide good quality spatial resolution, we hypothesized that proper image analysis could return quantitative information regarding both the thickness of the artery wall layers and the global architecture of the underlying connective tissue. Fig 3A and 3B show the main aspects of contrast-based ImageJ protocol along a linear profile drawn across the arterial wall of a negative TAB section (Fig 3A). The protocol was designed to access the most precise measurements for each tunica of the vessel. The gray-scale plot profile from Fig 3B confirms a significant rupture in contrast between the intima and media, as well as between the media and adventitia, allowing concomitant measurements of the thickness of each artery wall layer. Software provided by Hamamatsu facilitated similar measurements from scanned glass slides following HES staining, as exploited elsewhere [25]. In addition, Gabor filtering was applied to the same reconstructed negative TAB section in order to provide vector orientation maps and subsequent global analysis of the symmetry of the arterial section (Fig 3C). As expected from the previous qualitative analysis, Gabor filtering of FF-OCT-acquired negative TAB section returned a perfect orientation match from one point of the artery to its exact opposite following a 180-degree rotation, as demonstrated by the paired color system respecting an overall 180-degree rotational symmetry. A similar procedure was applied for the analysis of the gray-scale plot profile from positive TAB sections (Fig 3D). Due to a high heterogeneous contrast within the whole TAB section of GCA patients, Fig 3E shows almost no possible distinction between the different layers composing the artery wall with a contrast oscillating between 500 and 2000 arbitrary units from the very inner to the outer layer. Subsequent Gabor filtering of the positive section proves the pathological loss of the 180 degree rotational symmetry-based vector orientation match, as illustrated by the relative

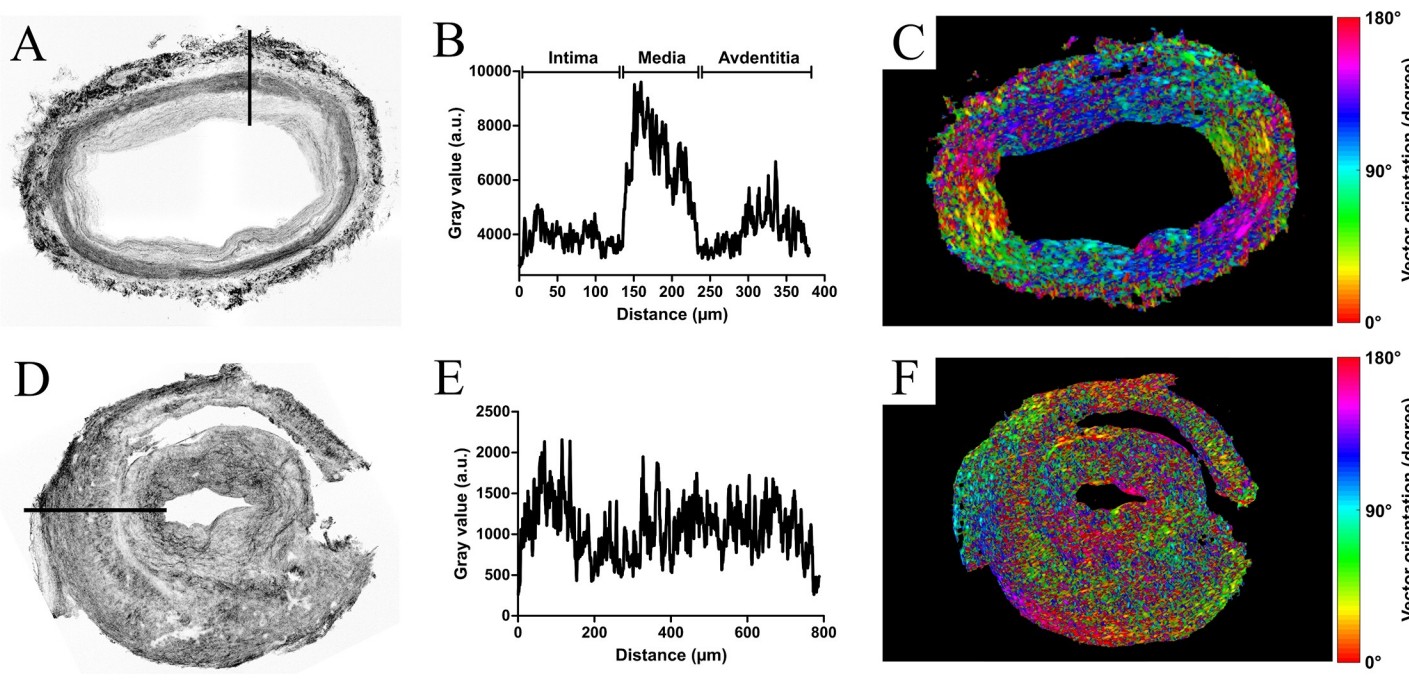

**Fig 3.**

chaotic color distribution within the layers of the artery resulting in a rainbow-like appearance (Fig 3F).

Fig 4A shows an example of a negative TAB for which FF-OCT images and conventional histology match perfectly from the inner to the outer layer of the biopsy. Regardless of intima thickness or GCA status (when quantifiable for GCA patients), quantitative analysis of both

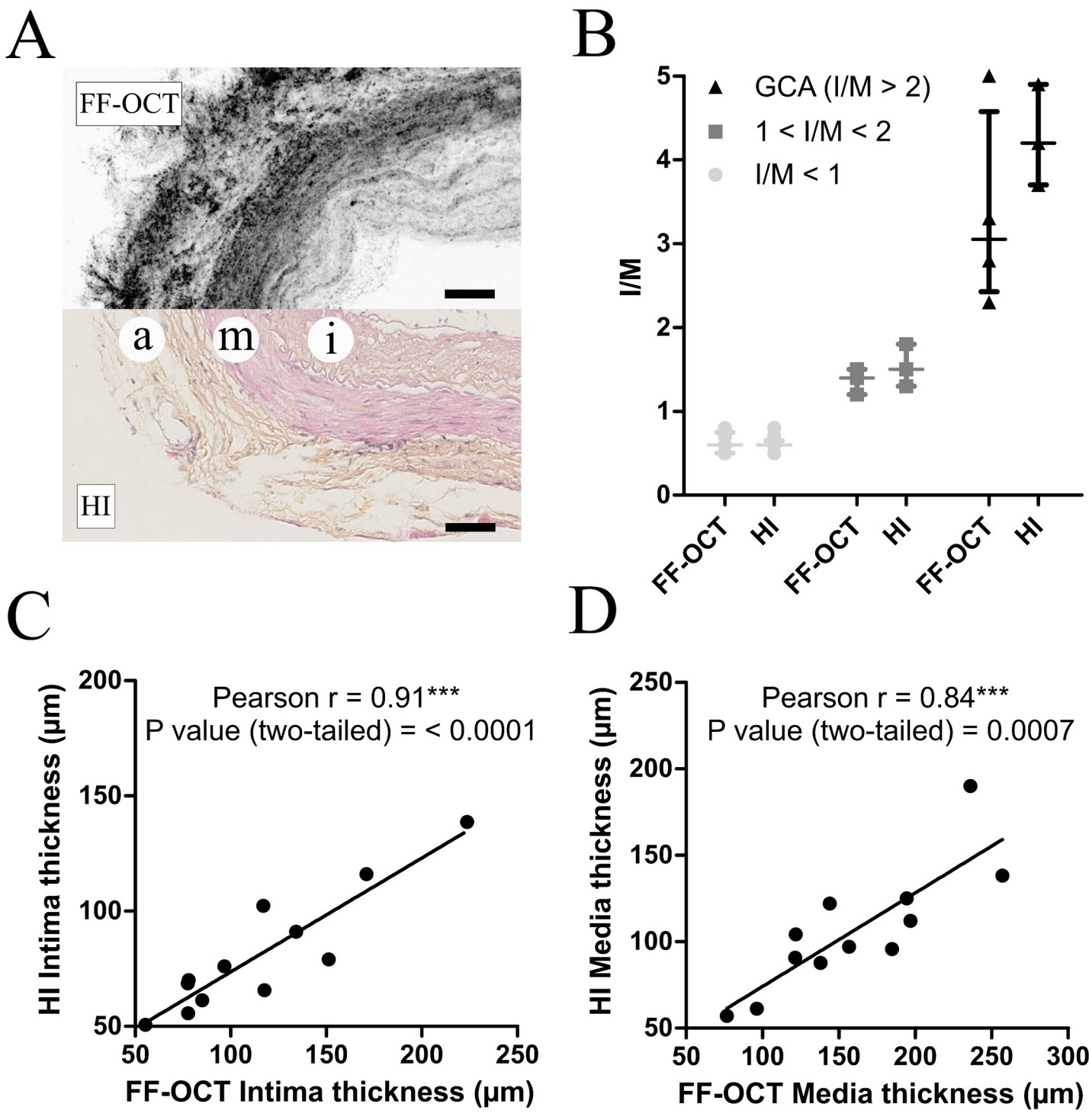

**Fig 4.**

intima and media thickness confirms the absence of statistical difference between FF-OCT-based and histology-based measurements of I/M (Fig 4B). Moreover, these results let to an accurate association of both quantitative classification and qualitative selection established by the pathologist for negative sections. Indeed, TABs with thin intima (S1 Fig) and TABs with intimal hyperplasia (S2 Fig) consequently appear as two separate groups: one with normal I/M <1 and another that shows an intimal hyperplastic response with I/M between 1 and 2. When data were accessible for TABs with GCA lesions, quantification brought out a third GCA group defined by I/M largely > 2. Finally, data from Fig 4C and 4D demonstrate a significant correlation between the thickness of the intima (Fig 4C) and media (Fig 4D) measured with FF-OCT and conventional histology.

## Discussion

The present work describes the first attempt to assess the potential of FF-OCT for the diagnosis of GCA in comparison with conventional histology. The first advantage of FF-OCT is that it provides rapid (within minutes) and on-site acquisition of TAB sections. We demonstrate, from the analysis of both healthy and GCA-positive TAB sections, that the high spatial resolution of FF-OCT technology makes it possible to visualize with precision several essential structures correlated with the diagnosis of GCA. Notably, we found that FF-OCT accurately returns both qualitative and quantitative information relative to the structure of the three arterial tissue layers and the IEL or *vasa vasorum*, with a significant correlation to histopathological imaging. When focusing on the FF-OCT analysis of healthy TAB sections, the inverted I/M can be interpreted as a reflection of the stage in human atherosclerosis [26]. Moreover, we provide preliminary proof that automated Gabor filtering could deliver both immediate and essential structural information regarding the preservation of the regular circularity of the media and intima-associated connective tissues, paving the way for potential machine intelligence-based pathological diagnosis of GCA. FF-OCT acquisitions return additional and complementary information with focus on the appearance and structural orientation of the underlying fibrous supporting tissue within each layer of the temporal artery. When TABs from GCA patients were compared to the global circular symmetry of healthy TAB sections, FF-OCT imaging revealed the destruction of the media layer and the modification of the arterial wall structure, which was rearranged into a denser reticular fibrous neo-intima, distinctive of GCA pathogenesis [27]. Despite the current success of non-invasive techniques like CDS, a precise FF-OCT-based analysis of the temporal artery wall on a meso-structural level remains of particular importance for the diagnosis of GCA. There is, however, a potential pitfall for GCA diagnosis with CDS since the atherosclerotic lesions responsible for significant increases in the thickness of the intima might mimic the halo sign, resulting in false positives [28].

We acknowledge several limitations that had an impact on the use of FF-OCT for rapid on-site pathological diagnosis of GCA in the current study. First, T-cells, macrophages and multi-nucleated cells, which are hallmarks of GCA, were not visible in the present set-up of FF-OCT, which used previously frozen TAB samples. Despite the high spatial resolution, the loss of information was due to the structural nature of contrast imaging, rendering direct black and white photographs of the specimen without any preparation or staining. However, this limitation is for the most part the result of using defrosted TAB samples with dead cellular material. This issue can be overcome by performing dynamic FF-OCT acquisition of fresh TAB sections, yielding complementary subcellular contrast [29] and putative direct visualization of inflammatory infiltrates. Moreover, FF-OCT-based image analysis allows z-stacking of a varying number of slices up to a cumulative length approaching 200 μm, highly dependent on the overall quality and sharpness of the initial TAB axial section, that may potentially improve

diagnosis accuracy by unveiling skip lesions. Such hypothesis would require additional experiments with a dedicated reproducible sampling technique allowing concomitant and iterative acquisition of the same slice with both FF-OCT and conventional histology throughout the whole length TAB section. Unfortunately, such experimental conditions remains to be found. Finally, the potential of *en face* white-light interference microscopy demonstrated in this work should encourage further investigations into the FF-OCT-based handheld acquisition probe [30], a promising technology dedicated to direct transcutaneous imaging and further non-invasive diagnosis of GCA.

## Conclusion

This preliminary study is the first to compare FF-OCT imaging to the gold standard histopathological procedure for the diagnosis of GCA. It brings conclusive proof regarding the potential of FF-OCT for both qualitative and quantitative structural visualization of underlying fibrous tissues and architectural changes in the arterial wall that occur throughout the inflammatory processes of GCA. After this first promising step, further investigations are warranted to confirm the potential of FF-OCT technology for rapid, on-site, non-invasive diagnosis of GCA.

## Supporting information

**S1 Fig. Qualitative imaging of healthy TAB specimens with thin intima layer (n = 9).** Comparison of FF-OCT (A, C, E, G, I, K, M, O, Q) and conventional histology (B, D, F, H, J, L, N, P, R) imaging. A and B correspond to niTAB1, C and D to niTAB2, E and F to niTAB3, G and H to niTAB4, I and J to niTAB5, K and L to niTAB5, M and N to niTAB7, O and P to niTAB8, Q and R to niTAB9.
(TIF)

**S2 Fig. Qualitative imaging of healthy TAB specimens with significant intimal hyperplasia (n = 3).** Comparison of FF-OCT (A, C, E) and conventional histology (B, D, F) imaging. A and B correspond to the ihTAB1, C and D to ihTAB2, E and F to ihTAB3.
(TIF)

**S3 Fig. Qualitative imaging of GCA TAB specimens (n = 4).** Comparison of FF-OCT (A, C, E, G) and conventional histology (B, D, F, H) imaging. A and B correspond to gcaTAB1, C and D to gcaTAB2, E and F to gcaTAB3, G and H to gcaTAB4.
(TIF)

## Acknowledgments

The authors thank Suzanne Rankin for her help in proofreading the article; Marion Ciudad, Marine Thébault, Claudie Cladière, Claire Gérard, Mathilde Charlot, Claire Boillin, Martine Breton, Amandine Esnoult, Thibault Ghesquière and Laetitia Barbier for their help in collecting samples; Céline Shaeffer, Eva Michaud and the Centre de Ressources Biologiques; Ferdinand Cabanne for their help in the conservation of frozen samples and Bertrand Le Conte de Poly for his help and advices in acquiring FF-OCT images at LLTech.

## Author Contributions

**Conceptualization:** Thomas Maldiney, Claude Boccara, Bertrand Tavitian, Sylvain Audia, Bernard Bonnotte, Maxime Samson.

**Data curation:** Thomas Maldiney, Hélène Greigert, Laurent Martin, Emilie Benoit, Jean-Marie Chassot, Claude Boccara, Daniel Balvay, Bertrand Tavitian, Olivier Clément, Bernard Bonnotte, Maxime Samson.

**Formal analysis:** Thomas Maldiney, Daniel Balvay, Bertrand Tavitian, Bernard Bonnotte, Maxime Samson.

**Funding acquisition:** Bernard Bonnotte, Maxime Samson.

**Investigation:** Thomas Maldiney, Jean-Marie Chassot, Bernard Bonnotte, Maxime Samson.

**Methodology:** Bernard Bonnotte, Maxime Samson.

**Project administration:** Bernard Bonnotte, Maxime Samson.

**Resources:** Bernard Bonnotte, Maxime Samson.

**Software:** Emilie Benoit.

**Supervision:** Bernard Bonnotte, Maxime Samson.

**Validation:** Thomas Maldiney, Catherine Creuzot-Garcher, Bernard Bonnotte, Maxime Samson.

**Visualization:** Pierre-Henry Gabrielle, Maxime Samson.

**Writing – original draft:** Thomas Maldiney, Bernard Bonnotte, Maxime Samson.

**Writing – review & editing:** Thomas Maldiney, Hélène Greigert, Laurent Martin, Emilie Benoit, Catherine Creuzot-Garcher, Pierre-Henry Gabrielle, Jean-Marie Chassot, Claude Boccara, Daniel Balvay, Bertrand Tavitian, Olivier Clément, Sylvain Audia, Bernard Bonnotte, Maxime Samson.

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
