## [Decision Letter · Decision Letter 0]

3 Aug 2020

PONE-D-20-14732

Full-field optical coherence tomography for the diagnosis of giant cell arteritis

PLOS ONE

Dear Dr. Maldiney,

Thank you for submitting your manuscript to PLOS ONE. After careful consideration, we feel that it has merit but does not fully meet PLOS ONE’s publication criteria as it currently stands. Therefore, we invite you to submit a revised version of the manuscript that addresses the points raised during the review process.

Please carefully address both reviewers' comments and make according revisions in the new version of your manuscript.

We look forward to receiving your revised manuscript.

Kind regards,

Stephan Meckel, MD, PhD

Academic Editor

PLOS ONE

Journal Requirements:

2. Please amend your Ethics statement by including approval numbers.

3.Thank you for stating the following in the Conflict of interest statement Section of your manuscript:

[All authors have no disclosure and no conflict of interest to declare. This work was supported by grants (MS) from the Fondation ARTHRITIS (2017-2018), the Groupement Interrégional Manuscript Click here to download Manuscript PLOS One Manuscript (Maldiney et al. 2020).docx FF-OCT for the diagnosis of GCA de Recherche Clinique et d’Innovation Est (GIRCI), Appel à Projet Jeunes Chercheurs 2013 and 2014 (Clinicaltrials.gov NCT02158208 and NCT02857192).]

 [The funders had no role in study design, data collection and analysis, decision to publish, or preparation of the manuscript.]

4.Thank you for stating the following in the Competing Interests section:

[The authors have declared that no competing interests exist.].   

We note that one or more of the authors are employed by a commercial company: LLTech SAS

Reviewers' comments:

Reviewer's Responses to Questions

**Comments to the Author**

1. Is the manuscript technically sound, and do the data support the conclusions?

Reviewer #1: Yes

Reviewer #2: Yes

2. Has the statistical analysis been performed appropriately and rigorously? 

Reviewer #1: Yes

Reviewer #2: Yes

3. Have the authors made all data underlying the findings in their manuscript fully available?

Reviewer #1: Yes

Reviewer #2: Yes

4. Is the manuscript presented in an intelligible fashion and written in standard English?

Reviewer #1: Yes

Reviewer #2: Yes

5. Review Comments to the Author

Reviewer #1: This is an interesting publication comparing FF-OCT and conventional histology for the diagnosis of GCA.

The authors convincingly demonstrate that FF_OCT accurately visualizes the main structure of the temporal

artery. The authors show a significant correlation of FF OCT and conventional histology.

I have few points:

- the included TABs were grouped in: niTAB, ihTAB and gcaTAB. In figure 2C/D the authors describe a TAB with

intimal hyperplasia and describe an increased intima thickniss compared to the TAB in figure 2 A/B.

- How was increased intima thickness defined.

- I find the difference in intima thickness between 2 A/B and C/D hard to see.

Furthermore, the authors did select the spot with the most pronounced intima thickness in figure 2 C/D. There were other

parts of the vessel wall, where the initima is much thiner.

Reviewer #2: The manuscript demonstrated the potential of using full field optical coherence tomography for diagnosis of giant cell arteritis. Images of temporal artery biopsy acquired with FF-OCT and images of conventional histology were compared, and the authors demonstrated the possibility of diagnosing GCA from FF-OCT images. The manuscript is well written, conclusions are supported by the results. The sample size in the study is relatively small, however, I understand it's difficult to increase the sample size significantly. The manuscript could also benefit if the authors briefly describe why FF-OCT, instead of other OCT modalities, e.g. SD-OCT, line scanning OCT, were used. Another minor thing is that the authors mentioned z stacks were acquired, but it was not shown in results, and whether acquiring z stacks improves diagnosis accuracy was not discussed.

6. PLOS authors have the option to publish the peer review history of their article (what does this mean?). If published, this will include your full peer review and any attached files.

Reviewer #1: No

Reviewer #2: No

---

## [Author Response · Author response to Decision Letter 0]

7 Aug 2020

Journal Requirements:

>> The Revised Manuscript was amended to meet PLOS ONE's style requirements according to the provided templates.

2. Please amend your Ethics statement by including approval numbers.

>> The approval numbers were included in the Ethics statement of the Revised Manuscript as follows :

[Patients included in this study participated in two studies involving GCA patients (Clinicaltrials.gov NCT02158208 and NCT02857192) and gave their oral consent for the use of their temporal arteries for subsequent research in the field of GCA. This study was approved by the Institutional Review Board and the Ethics Committee of the Dijon University Hospital.]

3.Thank you for stating the following in the Conflict of interest statement Section of your manuscript:

[All authors have no disclosure and no conflict of interest to declare. This work was supported by grants (MS) from the Fondation ARTHRITIS (2017-2018), the Groupement Interrégional Manuscript Click here to download Manuscript PLOS One Manuscript (Maldiney et al. 2020).docx FF-OCT for the diagnosis of GCA de Recherche Clinique et d’Innovation Est (GIRCI), Appel à Projet Jeunes Chercheurs 2013 and 2014 (Clinicaltrials.gov NCT02158208 and NCT02857192).]

Please remove any funding-related text from the manuscript and let us know how you would like to update your Funding Statement.

Currently, your Funding Statement reads as follows:

 [The funders had no role in study design, data collection and analysis, decision to publish, or preparation of the manuscript.]

>> As suggested, the funding-related text was removed from the Revised Manuscript and included in the cover letter as follows:

[This work was supported by grants from the 'Fondation ARTHRITIS (2017-2018)' and the 'Groupement Interrégional de Recherche Clinique et d’Innovation Est (GIRCI) - Appel à Projet Jeunes Chercheurs' 2013 and 2014 (Clinicaltrials.gov NCT02158208 and NCT02857192).

The funder did not provide any salary for authors, had no role in the study design, data collection and analysis, decision to publish, or preparation of the manuscript. The specific roles of these authors are articulated in the ‘author contributions’ section.

LLTech SAS did not play any role as a funding organization for this study. Besides, the commercial affiliation with LLTech SAS does not alter our adherence to PLOS ONE policies on sharing data and materials.]

4.Thank you for stating the following in the Competing Interests section:

[The authors have declared that no competing interests exist.].

We note that one or more of the authors are employed by a commercial company: LLTech SAS

>> The Funding Statement was updated to describe the role of LLTech SAS in the present study (see requirement 3).

Please also provide an updated Competing Interests Statement declaring this commercial affiliation along with any other relevant declarations relating to employment, consultancy, patents, products in development, or marketed products, etc.

>> The Competing Interests Statement was updated to describe the role of LLTech SAS in the present study and included in the cover letter as follows:

[We declare the paid employment of LLTech SAS (E. B.) to provide additional FF-OCT images acquisition and analysis.]

>> The sentence “data not shown” was removed from the Revised Manuscript. The following citation was added to support the sentence that refers to these data (l. 254-256):

[Gould PV, Saikali S. A Comparison of Digitized Frozen Section and Smear Preparations for Intraoperative Neurotelepathology. Anal Cell Pathol. 2012;35: 85–91. doi:10.1155/2012/454631].

Reviewers' comments:

Reviewer #1:This is an interesting publication comparing FF-OCT and conventional histology for the diagnosis of GCA. The authors convincingly demonstrate that FF-OCT accurately visualizes the main structure of the temporal artery. The authors show a significant correlation of FF OCT and conventional histology.

I have few points:

- the included TABs were grouped in: niTAB, ihTAB and gcaTAB. In figure 2C/D the authors describe a TAB with intimal hyperplasia and describe an increased intima thickness compared to the TAB in figure 2 A/B. How was increased intima thickness defined ?

>> The Revised Manuscript was modified in order to define the increased intima as follows (implemented in the "Histological TAB selection" subsection of "Materials and Methods" (l. 157-159):

[In these control samples, the pathologist studied the qualitative aspect of the temporal artery wall to distinguish between negative TAB with normal intima (thinner than media, referred to as niTAB.1 to 9, n = 9) and negative TAB with intimal hyperplasia (thicker than media, referred to as ihTAB.1 to 3, n = 3)].

- I find the difference in intima thickness between 2 A/B and C/D hard to see. Furthermore, the authors did select the spot with the most pronounced intima thickness in figure 2 C/D. There were other parts of the vessel wall, where the intima is much thinner.

>> We agree with the reviewer but we deliberately chose to highlight the most pronounced intima thickness in figure 2 C/D in order to better illustrate the difference between negative TAB with normal intima (i.e. thinner than media) and negative TAB with intimal hyperplasia. However, quantitative analysis and subsequent correlations took into account the other parts of the vessel wall - notably those for which intima was much thinner - as FF-OCT or scanned glass slides image-based tunica thickness were calculated as the mean of three representative measurements throughout each TAB section.

Reviewer #2:The manuscript demonstrated the potential of using full field optical coherence tomography for diagnosis of giant cell arteritis. Images of temporal artery biopsy acquired with FF-OCT and images of conventional histology were compared, and the authors demonstrated the possibility of diagnosing GCA from FF-OCT images. The manuscript is well written, conclusions are supported by the results. The sample size in the study is relatively small, however, I understand it's difficult to increase the sample size significantly.

The manuscript could also benefit if the authors briefly describe why FF-OCT, instead of other OCT modalities, e.g. SD-OCT, line scanning OCT, were used.

>> As suggested by the reviewer, the "Introduction" section of the Revised Manuscript was amended in order to briefly described the main advantages of FF-OCT when compared to other modalities. The following sentence, with additional reference, was implemented as follows (l. 126-129):

[When compared to other modalities such as conventional OCT or even confocal microscopy, FF-OCT was demonstrated to significantly improve spatial resolution by a factor varying from five to ten depending on the acquisition axis [21].].

Another minor thing is that the authors mentioned z stacks were acquired, but it was not shown in results, and whether acquiring z stacks improves diagnosis accuracy was not discussed.

>> The reviewer raises an important detail regarding the axial reconstruction of FF-OCT images and a precision was added at the end of the "FF-OCT imaging" subsection of "Materials and Methods" (l. 173-175):

[A series of FF-OCT slices with 1.5 µm spacing were recorded in depth, and ImageJ 1.52o software was used for axial reconstruction of TAB FF-OCT imaging following z-stacking of a minimum of 20 images.]

In addition, the improvement of diagnosis accuracy through z-stacking is discussed in the last part of the "Discussion" section as follows (l. 299-306): 

[Moreover, FF-OCT-based image analysis allows z-stacking of a varying number of slices up to a cumulative length approaching 200 µm, highly dependent on the overall quality and sharpness of the initial TAB axial section, that may potentially improve diagnosis accuracy by unveiling skip lesions. Such hypothesis would require additional experiments with a dedicated reproducible sampling technique allowing concomitant and iterative acquisition of the same slice with both FF-OCT and conventional histology throughout the whole length TAB section. Unfortunately, such experimental conditions remains to be found.]

---

## [Decision Letter · Decision Letter 1]

14 Aug 2020

Full-field optical coherence tomography for the diagnosis of giant cell arteritis

PONE-D-20-14732R1

Dear Dr. Maldiney,

We’re pleased to inform you that your manuscript has been judged scientifically suitable for publication and will be formally accepted for publication once it meets all outstanding technical requirements.

Kind regards,

Stephan Meckel, MD, PhD

Academic Editor

PLOS ONE

Additional Editor Comments (optional):

Reviewers' comments:

Reviewer's Responses to Questions

**Comments to the Author**

1. If the authors have adequately addressed your comments raised in a previous round of review and you feel that this manuscript is now acceptable for publication, you may indicate that here to bypass the “Comments to the Author” section, enter your conflict of interest statement in the “Confidential to Editor” section, and submit your "Accept" recommendation.

Reviewer #1: All comments have been addressed

Reviewer #2: All comments have been addressed

2. Is the manuscript technically sound, and do the data support the conclusions?

Reviewer #1: Yes

Reviewer #2: Yes

3. Has the statistical analysis been performed appropriately and rigorously? 

Reviewer #1: Yes

Reviewer #2: Yes

4. Have the authors made all data underlying the findings in their manuscript fully available?

Reviewer #1: Yes

Reviewer #2: Yes

5. Is the manuscript presented in an intelligible fashion and written in standard English?

Reviewer #1: Yes

Reviewer #2: Yes

6. Review Comments to the Author

Reviewer #1: (No Response)

Reviewer #2: The authors addressed all review comments effectively. I don't have further questions and believe the manuscript is ready for acceptance.

7. PLOS authors have the option to publish the peer review history of their article (what does this mean?). If published, this will include your full peer review and any attached files.

Reviewer #1: No

Reviewer #2: No

---

## [Editor Report · Acceptance letter]

18 Aug 2020

PONE-D-20-14732R1 

Full-field optical coherence tomography for the diagnosis of giant cell arteritis 

Dear Dr. Maldiney:

I'm pleased to inform you that your manuscript has been deemed suitable for publication in PLOS ONE. Congratulations! Your manuscript is now with our production department. 

Kind regards, 

on behalf of

Prof. Dr. Stephan Meckel 

Academic Editor

PLOS ONE